# AtomSurf-PPI: Protein-Protein Docking with Geometric Deep Learning Representations

**Yangyang Miao[1], Bruno Correia[1], Vincent Mallet[2,3,4]**

[1]Laboratory of Protein Design and Immunoengineering, Institute of Bioengineering,
Ecole polytechnique fédérale de Lausanne, Lausanne, Switzerland
[2]Mines Paris, PSL Research University, CBIO, Paris, France;
[3]Institut Curie, PSL Research University, Paris, France;
[4]INSERM, U1331, Paris, France;

`yangyang.miao@epfl.ch;vincent.mallet@minesparis.psl.eu`

## Abstract

Deep learning approaches to protein docking are fast, but do not yet reach the performance of traditional models. However, recent joint modeling of the surface and the graph of a protein enhance protein representations, notably for interaction prediction. In this paper, we show that embeddings learned by such models can efficiently guide classic point cloud alignment procedures and pose scoring models, resulting in a state-of-the-art protein-protein docking system.

Specifically, we propose **AtomSurf-PPI**, a multi-stage framework for protein-protein docking that integrates a dual-representation encoder, an enhanced Top-K RANSAC procedure for candidate pose generation, and a Graph Transformer-based scorer for final evaluation. AtomSurf-PPI consistently outperforms other deep learning methods and achieves large speedups over traditional search-based and co-folding methods. Code can be found online: `github.com/YangyangMiao/atomsurf_ppi`

## 1 Introduction

Proteins, the fundamental building blocks of life, often assemble into complexes to carry out intricate biological functions such as genome expression, immune responses, and more (Stelzl et al., 2005). It is estimated that over 80% of human proteins participate in $10^5$ to $10^6$ Protein-Protein Interactions (PPIs) (Stumpf et al., 2008; Yuan et al., 2025). PPI perturbation is frequently associated with diseases, such as genetic disorders and cancers (Li et al., 2017). Additionally, human pathogens often exploit interactions with host proteins to colonize cells (Brito & Pinney, 2017). Determining the three-dimensional (3D) structures of protein complexes can help elucidate the molecular mechanisms of these biological processes and develop therapeutic strategies.

Following the release of AlphaFold, to accurately predict monomer structures from sequence alignments (Jumper et al., 2021), several papers have proposed to directly fold two interacting chains together, a process termed co-folding. This includes papers that aimed to tweak the original architecture (Gao et al., 2022; Bryant et al., 2022; Ko & Lee, 2021), as well as others trained to natively accommodate co-folding (Evans et al., 2021; Abramson et al., 2024; Passaro et al., 2025). Since these approaches generate several candidates, scoring methods were also proposed (Evans et al., 2021; Zhu et al., 2023; Dunbrack Jr, 2025).

Co-folding has led to many interesting studies providing structural insights into protein-protein interactions at scale (Schmid & Walter, 2025; Zhang et al., 2025; Burke et al., 2023; Baptista et al., 2025). However, the quality of co-folding decreases on sequences far from the training set, or with shallow multiple sequence alignment (Škrinjar et al., 2025; Masters et al., 2025). Moreover, the computational cost of co-folding at scale can be daunting. For instance, Schmid & Walter (2025) report having used 500,000 GPU hours.

The more traditional approach for PPI structure prediction lies in computational docking. This approach uses a fast scoring function, a pose sampling algorithm and an optional refinement step. The fast scoring evaluates poses based on force fields (Shirali et al., 2025) or on statistical or semi-empirical functions that include heuristic terms, computed to fit some data (Thomas & Dill, 1996; Lu et al., 2003). A sampling algorithm explores the space of possible poses, for instance using geometric hashing (Schneidman-Duhovny et al., 2005) or fast Fourier transform for rigid docking (Yan et al., 2020), and molecular dynamics for flexible docking (Dominguez et al., 2003). Combining these procedures results in complex poses with a high score. This can be followed by a step of finer re-scoring to improve ranking estimates, often based on machine learning scoring functions (Wang et al., 2020; Réau et al., 2023). Recent pipelines propose to integrate flexible docking as a refinement step for co-folding approaches (Giulini et al., 2025). Traditional docking pipelines exhibit good generalization but long runtimes.

To accelerate rigid docking, machine learning methods have emerged to directly predict the relative orientations of two proteins. Pioneered by Ganea et al. (2021), they frame rigid docking as a regression problem (Ganea et al., 2021; Wang et al., 2023). In DiffDock, Corso et al. (2022) proposed to formulate the search for the pose as a generative problem, building on the insight that regression models could average several binding modes, resulting in a nonsensical solution. Originally developed for small molecule docking, it was adapted to PPI rigid docking (Ketata et al., 2023; Sverrisson et al., 2023), extended in the form of energy models (Wu et al., 2024; Chu et al., 2024) and formulated as a fine-tuning of structure prediction models (Xu et al., 2025).

Learning-based approaches have managed to drastically cut the computational time required to obtain a complex pose (10 seconds instead of 850), albeit at a significant cost in the performance. For instance, EBMDock (Wu et al., 2024) reports a mean DockQ of 0.05 versus 0.72 for HDock (Yan et al., 2020) on DB5.5 (Vreven et al., 2015). They always integrate a protein encoder with a generative model to sample poses.

*We believe learned protein embeddings are powerful assets to guide protein-protein docking, but challenge the idea that generative models are needed to generate convincing binding poses.*

In this paper, we propose AtomSurf-PPI, a geometric deep learning framework for protein-protein docking without a generative model. We train a joint graph and surface encoder (Mallet et al., 2025) to predict binding site and residue complementarity. We then generate candidate poses using an enhanced RANSAC algorithm that leverages learned surface embeddings to focus on binding sites and prioritize promising matches. Finally, we train a lightweight model to rescore the predicted poses.

AtomSurf-PPI manages to retain the computational efficiency of deep learning-based docking, while closing the performance gap to traditional methods (mean DockQ of 0.53 in 71 seconds on DB5.5). It also raises the deep docking performances on the challenging PINDER dataset (Kovtun et al., 2024) on Holo, Apo, and predicted structures. Furthermore, AtomSurf-PPI outperforms AF-Multimer on the Holo and Apo sets. This work demonstrates that meaningful embeddings can be used in conjunction with established registration algorithms to deliver state-of-the art results.

## 2 METHODS

We propose a multi-stage framework for protein-protein docking that integrates geometric deep learning with an efficient global search. Our method, built upon the AtomSurf architecture (Mallet et al., 2025), consists of three core components: (1) a dual-representation encoder for learning complementary surface-graph embeddings, (2) an embeddings-enhanced RANSAC procedure for candidate pose generation, and (3) a Graph Transformer-based ScoreNet for final docking pose ranking.

### 2.1 LEARNING SURFACE EMBEDDINGS RELEVANT TO DOCKING

**Encoding the surface** We aim to use RANSAC on the protein surface to perform docking, calling for surface embeddings. Hence, we learn protein embeddings using a recently proposed method (Mallet et al., 2025) that encodes protein structures jointly using a surface and a graph representation. Using multi-modal encoders results in powerful embeddings (Mallet et al., 2025; Zhang et al., 2024;

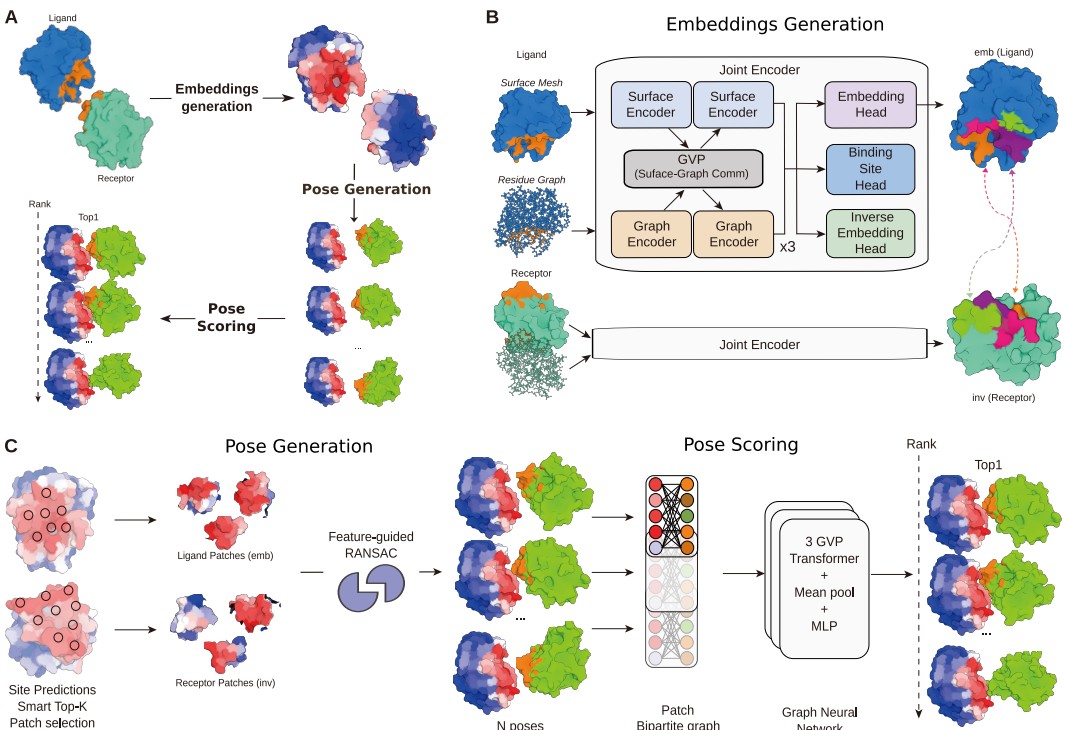

Figure 1: **A.** Overview of the AtomSurf-PPI Framework: generating protein embeddings, using those embeddings to generate poses and finally scoring and ranking these poses. **B.** Both the ligand and receptor proteins are represented with a graph and a surface, and independently encoded. The network is trained to predict binding sites, interacting pairs and interacting proteins across a batch. **C.** To efficiently generate poses, binding site predictions are used to extract non-overlapping patches, which are then paired and aligned with RANSAC using predicted pair compatibilities. Finally, these poses are scored by a lightweight graph neural network using pretrained features as input.

Qian et al., 2025; Zhou et al., 2025; Li et al., 2025). We propose a brief reminder of the architecture used and refer the interested reader to the paper for a more detailed description.

Let $\mathcal{S}_\mathbf{P}$ denote the surface of protein $P$, computed as a coarsened mesh generated with MSMS (Sanner et al., 1996), with vertices $\mathcal{V}_s$, faces $\mathcal{F}_s$ and features $\mathbf{X}_s$. Let $\mathcal{G}_P = (\mathcal{V}_g, \mathcal{E}_g)$ be a residue graph where nodes represent residues and edges are formed between 16 nearest neighbors, with features $\mathbf{X}_g$. Finally, a bipartite graph linking surface vertices and graph nodes is constructed and denoted as $G_{bp} = (V_{bp}, E_{bp})$ where $V_{bp} = \mathcal{V}_g \cup \mathcal{V}_s$.

AtomSurf relies on surface encoding layers $s_\theta^l$ and graph encoding layers $g_\theta^l$ to compute surface and graph intermediate embeddings, $\mathbf{H}_s^l = s_\theta^l(\mathbf{X}_s^l, \mathcal{F}_s)$ and $\mathbf{H}_g^l = g_\theta^l(\mathbf{X}_g^l, \mathcal{E}_g)$, where $L$ is the number of layers and $l \leq L$. These intermediate embeddings are then combined by another graph encoder $\mathfrak{g}_\theta^l$ acting on the bipartite graph, such that graph and surface representation are updated as $(\mathbf{X}_g^{l+1} || \mathbf{X}_s^{l+1}) = \mathfrak{g}_\theta^l(\mathbf{H}_g^l || \mathbf{H}_s^l, E_{bp})$.

The proposed surface encoder $s_\theta$ is a protein-adapted version of DiffusionNet Sharp et al. (2022), that captures surface geodesics and enables long distance message passing by leveraging heat diffusion. The residue graph encoder $g_\theta$ is ProNet Wang et al. (2022), which operates on residue graphs with features encoding the all-atom geometry of proteins. Finally, Geometric Vector Perceptrons Jing et al. (2021) are used for the message passing over the bipartite graph $\mathfrak{g}_\theta$, ensuring geometry-aware cross-modality integration.

On top of the AtomSurf encoder, we add several projection heads to empower docking with the learned features $\mathbf{X}^L$. Each head is implemented as a two layer MLP. First, we use a projection head $p_{bs}$ to predict binding sites, resulting in predictions $\mathbf{x}_{bs} = p_{bs}(\mathbf{X}^L)$. In addition, to enable effective feature-based alignment during docking, we want to efficiently compare pairs of residues. To do

so, we need to learn complementary rather than identical representations for two points in contact, corresponding to convex-concave complementarity Gainza et al. (2020). Therefore, we also use two independent heads $p_{\mathrm{emb}}$ and $p_{\mathrm{inv}}$, to project our features into distinct embedding spaces, resulting in final embeddings $\mathbf{x}_{\mathrm{emb}}$ and $\mathbf{x}_{\mathrm{inv}}$.

**Training the model**  The model is trained on the PINDER dataset Kovtun et al. (2024). This large-scale dataset is built using experimental and predicted complex structures. One of its key features is the careful splitting procedure, based on interface similarity, preventing data leakage. We use the clustered version of the dataset that holds 42k complexes, and train our networks only with holo structures to evaluate the robustness of our method to conformational changes or prediction errors.

Let us recall the expressions of focal loss Lin et al. (2017) and margin losses,

$$\mathcal{L}_{\mathrm{focal}}(p,y,\gamma) = -y(1-p)^{\gamma}\log(p) - (1-y)p^{\gamma}\log(1-p), \tag{1}$$

$$\mathcal{L}_{\mathrm{margin}}(e_i,e_j,y,d,m) = y \cdot d(e_i,e_j) + (1-y) \cdot \max(0, m - d(e_i,e_j)), \tag{2}$$

where $y$ is a binary target, $p$ is a predicted probability, $e_i$ and $e_j$ are vectors of the Euclidean space, $d$ is a distance function, and $\gamma$ and $m$ are scalar parameters. The focal loss is a modified version of binary cross entropy focusing on hard examples, used with $\gamma = 2$ in our experiments. The margin loss collapses positive pairs in space and pushes negative ones above a certain margin $m$.

Let us now detail the loss used to train our networks. Put shortly, We include a component to supervise our binding site head and our pair-wise predictions. We add a loss at the whole protein-level as an extra supervision. We supervise both our graph and our surface representations.

Given a protein dimer, we consider pairs including elements of each monomer. A pair of graph nodes is defined as positive if it contains atoms closer than 5 Å. We use all pairs of the graph. On the surface, vertices are defined to be positive if they are closer than 2Å. We use all positive pairs and ten times more negative pairs. This results in a total of $n$ pairs $(i_k, j_k)$ with labels $y_k, k \leq n$. We define as positive nodes and vertices the ones involved in at least one positive pair, and use the same protocol for negatives, resulting in $m$ points with labels $y_i^{\mathrm{site}}, i \leq m$.

To provide the model with a coarse-grained understanding of where interactions occur, we first supervise the binding site head using a simple focal loss, $\mathcal{L}_{\mathrm{site}} = \lambda_{\mathrm{site}} \sum_{i \leq m} \mathcal{L}_{\mathrm{focal}}(\mathbf{x}_{bs,i}, y_i)$, where $\lambda_{\mathrm{site}} = 4$.

In addition, to capture the lock-and-key nature of PPIs, we optimize for pair feature complementarity. Let $\mathcal{L}_{cos}$ be the margin loss used with the distance function $d(x,y) = 1 - cos(x,y)$ and $m_{\mathrm{cos}} = 1.5$ and $\mathcal{L}_{\mathrm{euc}}$ the one used with the $L_2$ distance function and $m_{\mathrm{euc}} = 2$. $\mathcal{L}_{\mathrm{pair}}$ combines these two margin losses and a focal loss following Equation 3 with $\lambda_{\mathrm{pair}} = 5$. Finally, our total complementarity loss $\mathcal{L}_{\mathrm{comp}}$ is computed over our $n$ pairs and symmetrized in Equation 4.

$$\mathcal{L}_{\mathrm{pair}}(e_i,e_j,y) = \mathcal{L}_{\mathrm{focal}}(\langle e_i,e_j \rangle, y) + \lambda_{\mathrm{pair}}\mathcal{L}_{\mathrm{cos}}(e_i,e_j,y) + \lambda_{\mathrm{pair}}\mathcal{L}_{\mathrm{euc}}(e_i,e_j,y), \tag{3}$$

$$\mathcal{L}_{\mathrm{comp}} = \sum_{k \leq n} \mathcal{L}_{\mathrm{pair}}(\mathbf{x}_{\mathrm{emb},i_k}, \mathbf{x}_{\mathrm{inv},j_k}, y_k) + \mathcal{L}_{\mathrm{pair}}(\mathbf{x}_{\mathrm{inv},i_k}, \mathbf{x}_{\mathrm{emb},j_k}, y_k). \tag{4}$$

Finally, we supervise protein-level interaction prediction. To get the representation $\mathbf{x}_u$ of a protein $u$, we average pool its residue representations. Non-interacting proteins are generated by shuffling the pairs within a batch, and added to the positive pairs, resulting in $q$ labeled pairs of proteins $(u_k, v_k, y_k), k \leq q$. We use the same supervision as for node pairs, $\mathcal{L}_{\mathrm{prot}} = \sum_{k \leq q} \mathcal{L}_{\mathrm{pair}}(\mathbf{x}_{u_k}, \mathbf{x}_{v_k}, y_k)$

Our final training loss is obtained by summing these three contributions: $\mathcal{L}_{\mathrm{site}} + \mathcal{L}_{\mathrm{comp}} + \mathcal{L}_{\mathrm{prot}}$.

In the original AtomSurf paper, the loss only included residue-level supervision for two focal losses; one for pairs and one for binding site prediction.

## 2.2 FEATURES-GUIDED RANSAC DOCKING

RANdom SAmple Consensus (RANSAC) methods aim to solve a data-related problem by iteratively solving the same problem on sub-samples of the data, and keeping the solution giving the best score on the overall data. In registration problems, it finds a global registration that aligns few points and computes a global score.

Our overall pipeline can be described as a bi-level RANSAC procedure, guided by our learned features. To globally align proteins, we first select large sub-samples (patches) upon which we compute local alignments and score the corresponding global alignments with ScoreNet (section 2.3). The procedure used to compute local alignment is also a RANSAC algorithm, as detailed below.

**Subsample selection with binding site predictions** The first use of our learned features is to efficiently select the sub-samples used by the outer RANSAC. To do so, we sort our surface points by predicted binding site probability. However, simply selecting the top-scoring vertices would lead to redundant geometric coverage in high-confidence regions. Hence, we introduce a Smart Top-K strategy that enforces selected vertices to be at least 5Å away from all previously selected centers. Finally, we extract mesh patches around each selected center based on geodesic neighborhoods (radius = 12Å).

Without binding site prediction, the exhaustive docking is computationally prohibitive. Adding the distance constraint achieves a 68% reduction in total alignment tests while improving mean RMSD by 0.17Å compared to unconstrained selection.

**Local alignments with complementary feature matching** Equipped with a pair of patches $R = (i_1, i_2, ..i_m)$ and $L = (j_1, j_2, ..j_n)$, we align them using a function from the Open3D library (`registration_ransac_based_on_feature_matching`). The algorithm selects a minimal set of 4 random vertices from $L$ and identifies their corresponding points in $R$ using the precomputed features. To do so, it identifies $\text{neigh}(j) = \text{argmin}_{i \in R} \langle x^R_{\text{emb},i}, x^L_{\text{inv},j} \rangle$ for each of the 4 sampled points. By calculating the nearest neighbors in this learned feature space, we ensure that interacting surface regions exhibit high feature similarity, which facilitates robust correspondence identification. In addition to feature matching, the algorithm also uses checks based on edge lengths, distances, and surface normals.

Once four pairs of points across patches have been identified, the algorithm finds the roto-translation that best aligns these points, and computes the number of inlier correspondences within a distance threshold of 2Å. These steps are repeated up to $900,000$ times or before if it satisfies a desired confidence level (set to $0.999$).

After RANSAC provides a coarse local alignment, the transformation is further optimized using the Iterative Closest Point (ICP) algorithm. Specifically, we employ a Point-to-Plane ICP refinement step. This minimizes the error between the ligand patch vertices and the local planar surfaces of the receptor, resulting in a tighter, more precise final docking pose.

## 2.3 ScoreNet: Pose Evaluation and Ranking

To identify the optimal docking poses from the RANSAC-generated candidate complex poses, we developed ScoreNet, a lightweight Graph Transformer-based scoring function. The network processes a pair of aligned surface patches to predict a continuous quality score $s \in [0, 1]$ reflecting the alignment precision.

To train our network, we produce a large number of alignments on our train and validation sets, and compute the RMSD to the ground truth complex. These RMSD values are mapped to quality scores $q$ using a sigmoid transformation: $q(R, L) = \text{sigmoid}(1.5(5.0 - \text{RMSD}(R, L)))$. This ensures that near-native alignments (RMSD$< 2$Å) receive scores near 1.0, while incorrect poses (RMSD$> 10$Å) are penalized with scores near 0.0.

We model inter-patch relationships by constructing a bipartite graph in the aligned 3D space, where each vertex in the ligand patch is connected to its five nearest neighbors in the receptor patch. Edges incorporate three-dimensional features: (1) Euclidean distance, (2) normal dot product for orientation compatibility, and (3) cosine similarity of the learned embeddings to evaluate complementarity.

ScoreNet utilizes three stacked Graph Transformer (Dwivedi & Bresson, 2020) layers with 4-head attention. These layers integrate the multi-scale edge features to weigh message passing based on local interaction quality. Global mean pooling aggregates these features into a graph-level representation, which an MLP head then processes to output the final score.

The model is optimized using a combined MSE and Smooth L1 loss. To address the scarcity of high-quality poses, we employ a balanced sampling strategy with a 1:1:2 ratio for positive, medium, and negative samples. This approach yields a high predictive reliability, achieving a Pearson correlation of 0.912 on the test set.

# 3 Experimental Setting

## 3.1 Datasets

We evaluate AtomSurf-PPI on two prominent benchmarks for protein-protein docking. Docking Benchmark 5.5 (DB5.5) (Vreven et al., 2015) is a gold-standard dataset containing 230 non-redundant, high-quality structures of protein-protein complexes. It is manually curated by experts and includes both bound (Holo) and unbound (Apo) structures for each component.

We train on PINDER (Kovtun et al., 2024), but also test on their official test set. PINDER was introduced to provide a dataset with orders of magnitude more data and a stricter data splitting procedure for protein docking research. Notably, PINDER is the first dataset to include paired predicted and Apo structures at scale, enabling the training of flexible docking methods.

## 3.2 Baselines

We compare AtomSurf-PPI against a diverse set of state-of-the-art docking methods. First, we include traditional non-machine learning approaches, HDock (Yan et al., 2020) and PatchDock (Schneidman-Duhovny et al., 2005), which rely on geometric complementarity and physics-based scoring functions. While these methods usually provide strong performances, they have a high computational cost.

We also include AlphaFold-Multimer (AF2-Multimer) (Evans et al., 2021), a co-folding method which predicts complex structures directly from multiple sequence alignments via a multi-track transformer. Note that co-folding methods do not require monomer structures. They are also the only method of this benchmark to allow for flexible docking, i.e. the protein is not assumed to be rigid upon binding.

Finally, we compare to other deep learning-based algorithms. DiffDock-PP (Ketata et al., 2023) is a generative model that treats docking as a diffusion process over SE(3) transformations. EBMDock (Wu et al., 2024) is a recent energy-based generative model that also employs its statistical potential for pose optimization. These models have the fastest runtimes, but until now, deep learning-based methods tend to underperform other approaches.

## 3.3 Evaluation Metrics

To comprehensively assess docking quality, we employ four standard metrics:

- $L_{\mathrm{RMS}}$ (Ligand RMSD): The RMSD of the backbone atoms of the ligand after superimposing the predicted and native receptor structures.

- $C_{\mathrm{RMS}}$ (Complex RMSD): The RMSD of the backbone atoms of the complex after superimposing the predicted and native receptor structures.

- $i_{\mathrm{RMS}}$ (Interface RMSD): The RMSD over backbone atoms of interface residues between the predicted model and the target structure. Interface residues are defined using a 10Å atomic contact cutoff.

- $F_{nat}$ (Fraction of Native Contacts): The proportion of native interfacial contacts preserved in the predicted complex. Contacts are defined as pairs of heavy atoms from the receptor and ligand within a 5Å distance.

In addition, we use integrated metrics that combine these values. First, we use the traditional CAPRI classification which has been used by the competition for over 20 years. It splits predictions in four quality bins:

$$\text{High} := \{F_{\text{nat}} \geq 0.5 \ \wedge \ (L_{\text{RMS}} \leq 1.0 \ \vee \ i_{\text{RMS}} \leq 1.0)\}$$
$$\text{Medium} := \{0.3 \leq F_{\text{nat}} < 0.5 \ \wedge \ (L_{\text{RMS}} \leq 5.0 \ \vee \ i_{\text{RMS}} \leq 2.0)\}$$
$$\text{Acceptable} := \{0.1 \leq F_{\text{nat}} < 0.3 \ \wedge \ (L_{\text{RMS}} \leq 10.0 \ \vee \ i_{\text{RMS}} \leq 4.0)\}$$
$$\text{Incorrect} := \{F_{\text{nat}} < 0.1 \ \vee \ (L_{\text{RMS}} > 10.0 \ \wedge \ i_{\text{RMS}} > 4.0)\}$$

DockQ (Basu & Wallner, 2016) was introduced to turn the discrete classification of CAPRI into a single continuous quality score in the range $[0, 1]$ that integrates $F_{nat}$, $LRMS$, and $iRMS$ into a single value. It is defined as:

$$f(L_{\text{RMS}}, i_{\text{RMS}}, F_{nat}) = \frac{F_{nat} + \frac{1}{1+(i_{\text{RMS}}/d_2)^2} + \frac{1}{1+(L_{\text{RMS}}/d_1)^2}}{3},$$

where the scaling factors are set to $d_1 = 8.5\text{Å}$ and $d_2 = 1.5\text{Å}$.

For holo-structure docking, predicted poses are compared directly to the ground-truth holo complex. For Apo and predicted structures, we follow the PINDER protocol by aligning coordinates to the holo frame to ensure consistent RMSD calculation. Here we perform sequence-guided structural alignment. We extract $C\alpha$ atoms from Apo or predicted and holo structures, perform global sequence alignment using BLOSUM62, and retain aligned residue pairs with positive scores as initial anchors. An iterative Kabsch superimposition with IQR-based (Interquartile Range) outlier rejection (removing points beyond $1.5 \times IQR$ then computes the optimal rotation R and translation t that maps the AF coordinates to the holo frame. This robust alignment accounts for sequence variations and structural differences, enabling accurate residue-wise matching for subsequent docking evaluation.

## 4 RESULTS

**DB5.5 Results**. The Docking Benchmark Dataset 5.5 results are shown in Table 1. Among physics-based methods, HDock outperforms PatchDock across the board, with an average DockQ score of 0.72. On this set, AF2-Multimer does not have as good results as physics-based methods (mean DockQ 0.48) and displays a longer runtime (2500s vs 850s). Finally, among deep learning-based baselines EBMDock offers a better performance than DiffDock-PP, at a fraction of the computational time of co-folding or physics-based approaches (10s). However, the performance of these approaches falls far behind (Mean DockQ 0.05).

AtomSurf-PPI outperforms the other deep learning-based baselines across the benchmark dataset, notably displaying acceptable DockQ values (0.53 on average). This performance is comparable to co-folding and below the one of HDock. However, AtomSurf-PPI offers a much lower computational cost (71s), representing an interesting balance between performance and throughput.

The results show that the Top-5 predictions from AtomSurf-PPI with ScoreNet yield substantially lower RMSD and higher DockQ than its Top-1 output, indicating the limitations of our scoring network. This is further demonstrated by our Oracle version, which selects the best among sampled poses. This represents the upper bound of our sampling method, achieving notably low RMSD and high DockQ.

**PINDER Results**. We evaluate AtomSurf-PPI on the comprehensive PINDER-XL set (Table 3). We also validate on its subset, PINDER-AF2 (Table 2), designed to test generalization for AlphaFold-multimer.

On both datasets, across settings, AtomSurf-PPI consistently and significantly outperforms both classical (PatchDock) and recent deep learning (DiffDock-PP) docking methods across Holo and Apo input structures in Top-1, Top-5, and Oracle success rates. However, its performance falls short of the more expansive, physics-based HDock that represents a strong baseline.

Comparison on the PINDER-AF2 set, allows us to compare AF2-Multimer with other tools. In the Holo and Apo settings, as observed in the previous experiments, its performance falls short of HDock's. In these settings, AtomSurf-PPI also outperforms AF2-Multimer even in the Top1 setting and in the Apo setting. This is a strong result for AtomSurf-PPI that demonstrates its robustness beyond temporal splits and conformational changes.

Table 1: Docking performance of the different methods on the DB5.5 dataset. We include results when the best results is computed from our five highest scoring outputs (Top-5) and from all our constructs, bypassing the scoring network (Oracle).

| Method | Complex RMSD ↓ | | | Interface RMSD ↓ | | | DockQ ↑ | | | Time |
|---|---|---|---|---|---|---|---|---|---|---|
| | Mean | Med. | Std. | Mean | Med. | Std. | Mean | Med. | Std. | (s) |
| HDock | 5.55 | 0.42 | 9.42 | 5.19 | 0.31 | 8.99 | 0.72 | 0.97 | 0.42 | 850 |
| PatchDock | 19.34 | 17.95 | 10.30 | 17.16 | 16.17 | 10.35 | 0.04 | 0.02 | 0.20 | 2232 |
| AF2-Multimer | 7.65 | 4.86 | 8.03 | 6.41 | 1.69 | 7.66 | 0.48 | 0.55 | 0.10 | 2503 |
| DIFFDOCK-PP | 17.56 | 17.21 | 7.87 | 17.76 | 17.12 | 8.47 | 0.04 | 0.01 | 0.09 | 37 |
| EBMDock | 14.79 | 15.64 | 5.05 | 12.47 | 10.99 | 5.06 | 0.05 | 0.04 | 0.04 | 10 |
| AtomSurf-PPI ScoreNet Top-1 | 10.12 | 9.54 | 8.01 | 8.99 | 8.80 | 7.76 | 0.47 | 0.34 | 0.21 | 71 |
| AtomSurf-PPI ScoreNet Top-5 | 6.80 | 4.29 | 7.10 | 6.27 | 3.93 | 6.81 | 0.53 | 0.40 | 0.22 | 71 |
| AtomSurf-PPI Oracle | *3.04* | *1.86* | *3.44* | *2.94* | *1.46* | *3.46* | *0.63* | *0.65* | *0.22* | 71 |

Table 2: Performance comparison of the different methods on PINDER-AF test set

| Input | Top1 | | | Top5 | | | Oracle | | |
|---|---|---|---|---|---|---|---|---|---|
| Method | Acceptable | Medium | High | Acceptable | Medium | High | Acceptable | Medium | High |
| **Holo Structures** | | | | | | | | | |
| HDOCK | 86.11 | 85.56 | 85.0 | 89.44 | 88.89 | 88.33 | 92.78 | 91.67 | 90.0 |
| PatchDock | 46.11 | 45.56 | 36.11 | 55.0 | 53.33 | 43.33 | 67.78 | 62.22 | 47.22 |
| AF2-Multimer | 54.44 | 48.89 | 23.89 | 57.78 | 51.11 | 28.33 | 57.78 | 51.11 | 28.33 |
| DiffDock-PP | * | * | * | * | * | * | 58.33 | 37.78 | 15.56 |
| AtomSurfPPI | 67.7 | 66.6 | 58.0 | 77.5 | 74.1 | 64.9 | 86.7 | 78.7 | 68.4 |
| **Apo Structures** | | | | | | | | | |
| HDOCK | 45.32 | 42.4 | 35.38 | 51.46 | 47.66 | 38.3 | 61.11 | 53.51 | 40.94 |
| PatchDock | 36.67 | 30.0 | 20.0 | 36.67 | 36.67 | 23.33 | 56.67 | 46.67 | 26.67 |
| AF2-Multimer | 20.0 | 16.67 | 16.67 | 20.0 | 16.67 | 16.67 | 20.0 | 16.67 | 16.67 |
| DiffDock-PP | * | * | * | * | * | * | 30.0 | 10.0 | 0.0 |
| AtomSurfPPI | 31.2 | 25.0 | 18.8 | 31.3 | 31.3 | 18.8 | 50.1 | 31.3 | 18.8 |
| **Predicted Structures** | | | | | | | | | |
| HDOCK | 16.54 | 12.6 | 4.72 | 18.9 | 14.17 | 4.72 | 26.77 | 17.32 | 4.72 |
| PatchDock | 7.87 | 6.3 | 1.57 | 12.6 | 11.02 | 2.36 | 16.54 | 13.39 | 2.36 |
| AF2-Multimer | 57.48 | 51.97 | 26.77 | 59.84 | 54.33 | 32.28 | 59.84 | 54.33 | 32.28 |
| DiffDock-PP | * | * | * | * | * | * | 30.71 | 12.6 | 0.79 |
| AtomSurfPPI | 2.4 | 1.6 | 0.8 | 4.80 | 1.5 | 0.80 | 35.50 | 10.50 | 0.80 |
| *Rigid Oracle* | * | * | * | * | * | * | *92.3* | *85.7* | *55.6* |

On predicted structures, the performance of all methods but AF2-multimer collapse. We investigated if there was a hard bound on the performance of rigid docking. To do so, we introduce a *Rigid Oracle* method where we use ground truth binding site and pair correspondences in our method. This shows that good solutions are achievable with rigid docking approaches.

Hence, this performance drop likely reflects a strong distributional shift. In this setting, even HDock is outperformed by AF2-Multimer. Note that we explicitly trained on holo structures to explore the robustness of our architecture. However, despite high global folding precision, AF2-predicted monomers often harbor small errors in surface residue orientations. These conformational artifacts deviate from the high-fidelity interfacial features used during training, ultimately hindering docking accuracy. We also note a strong performance drop for our method between the Oracle and the Top-5 setting. This indicates that the scoring network is also strongly affected by this distributional shift.

## 5 DISCUSSION

In this paper, we have explored the possibility of using deep learning to extract relevant features in conjunction with classical registration algorithms for rigid protein docking. Deep feature-infused docking models consistently and significantly outperformed ones that also try to find the optimal alignment with generative models, offering a promising alternative and challenging current research. Our approach delivers a solid option, balancing between robust performance and rapid execution.

Our work includes several limitations. First and foremost, its performance still lags behind HDock which we attribute to missed binding site detections, while HDock performs exhaustive search. Moreover, the performance of our approach drastically drops on predicted monomer structures. We explicitly refrained from training on such structures to evaluate the robustness of our embeddings to predicted structures. Future work could fine-tune our models specifically for predicted models. More generally, we would be interested in applying our model to other PPI-related tasks, such as PPI detection.

## MEANINGFULNESS STATEMENT

Our paper consists in investigating the usefulness of learned embeddings for protein protein docking perfectly aligning with the objectives of the workshop.

## ACKNOWLEDGMENTS

V.M. is supported by a Junior Springboard Prairie program, funded by the ANR project ANR-23-IACL-0008. This work was performed using HPC resources from GENCI–IDRIS (Grant AD010317140). Y.M and B.C. are supported by Swiss National Science Foundation grants 310030_197724, TMGC-3_213750 and 200020_214843. This work was performed using HPC resources from GENCI–IDRIS (Grant 2023-AD010613356) and CITAS at EPFL.

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

## APPENDIX

Table 3: Performance comparison of the different methods on PINDER-XL test set

| Input Method | Top1 Acceptable | Medium | High | Top5 Acceptable | Medium | High | Oracle Acceptable | Medium | High |
|---|---|---|---|---|---|---|---|---|---|
| **Holo Structures** | | | | | | | | | |
| HDOCK | 91.25 | 90.9 | 90.18 | 94.68 | 94.32 | 93.35 | 96.42 | 95.91 | 94.78 |
| PatchDock | 56.68 | 55.19 | 46.6 | 65.52 | 63.53 | 52.63 | 75.86 | 71.76 | 58.52 |
| DiffDock-PP | * | * | * | * | * | * | 70.74 | 50.95 | 23.38 |
| AtomSurfPPI | 69.5 | 66.2 | 57.7 | 77.2 | 72.2 | 64.3 | 85.4 | 76.1 | 67.3 |
| **Apo Structures** | | | | | | | | | |
| HDOCK | 45.32 | 42.4 | 35.38 | 51.46 | 47.66 | 38.3 | 61.11 | 53.51 | 40.94 |
| PatchDock | 19.59 | 17.25 | 11.99 | 26.32 | 22.22 | 12.57 | 37.13 | 30.12 | 14.04 |
| DiffDock-PP | * | * | * | * | * | * | 41.52 | 21.05 | 4.39 |
| AtomSurfPPI | 30 | 24.6 | 15.3 | 41.1 | 30.9 | 16.8 | 52.2 | 31.8 | 17.1 |
| **Predicted Structures** | | | | | | | | | |
| HDOCK | 48.25 | 44.02 | 26.33 | 53.86 | 49.17 | 28.85 | 58.79 | 52.89 | 29.59 |
| PatchDock | 28.05 | 25.3 | 10.07 | 34.97 | 31.08 | 12.25 | 44.42 | 38.29 | 13.74 |
| DiffDock-PP | * | * | * | * | * | * | 50.09 | 33.49 | 6.47 |
| AtomSurfPPI | 4.53 | 3.44 | 1.09 | 10.01 | 6.07 | 1.83 | 49.75 | 22.75 | 5.05 |

