# OpenReview forum: "AtomSurf-PPI: Protein-Protein Docking with Geometric Deep Learning Representations"
_ICLR.cc/2026/Workshop/LMRL — ICLR 2026 Workshop LMRL Poster_

### Official Review · Reviewer_8STq · 2026-02-11
**A Hybrid Geometric Deep Learning Framework for Efficient Protein–Protein Docking**

**Rating:** 7
**Confidence:** 4

**Review:**

This paper proposes AtomSurf-PPI, a multi-stage framework for rigid protein–protein docking. The method leverages the AtomSurf dual surface–graph encoder to learn binding site predictions and complementary embeddings, which are then used to guide a feature-enhanced RANSAC alignment procedure. Finally, a ScoreNet ranks candidate docking poses.

Overall, the paper presents an interesting and well-motivated application of learned geometric representations to protein–protein docking. The experimental results demonstrate competitive performance compared to other deep learning approaches, while offering substantial computational savings. The method represents a promising direction for bridging learned representations and established algorithms.

That said, several aspects of the paper would benefit from improved clarity and exposition.

## 1. Clarity of notation and definitions

In Section 2.1 (around line 157), the variables $x_{bs}, x_{emb}, x_{inv}$ are introduced as outputs of the projection heads, but they are not clearly defined beforehand. While the projection heads are mentioned, it would improve readability to explicitly state that:
	•	$x_{bs}$ corresponds to binding site predictions,
	•	$x_{emb}$ and $x_{inv}$ are complementary embeddings used for feature matching.
A short clarifying sentence when these variables are first introduced would significantly improve the flow of the method section.


## 2. Separation between prior AtomSurf components and novel contributions

It is sometimes unclear which parts of the architecture and training losses are inherited directly from AtomSurf and which are novel contributions of this paper.
For example:
- Is the complementarity loss (Eq. 3–4) newly introduced for docking, or part of the original AtomSurf framework?
- Is the protein-level interaction loss newly added?
- Are all projection heads specific to this work?
A clearer distinction between reused components and new contributions would help readers better assess the novelty of the approach.


## 3. Explanation of the different AtomSurf-PPI variants

In the results tables (e.g., DB5.5 and PINDER results), multiple variants are reported: ScoreNet Top-1, ScoreNet Top-5, Oracle.

However, the exact procedural differences between these variants are not fully explained in the main text. A concise explanation in the experimental section would make the evaluation protocol much clearer.


## 4. Presentation of results tables

The results tables would benefit from improved readability: adding boldface for the best results (and possibly underline for second-best) would make comparisons much easier.
Given that the paper emphasizes competitiveness and efficiency trade-offs, clearer visual guidance in the tables would strengthen the empirical claims.


# Overall Assessment

The paper presents a compelling hybrid approach that integrates learned geometric representations with classical docking algorithms. The idea of using complementary surface embeddings to guide RANSAC alignment is novel and well motivated. The empirical results show a strong improvement over existing deep learning docking methods while maintaining favorable computational efficiency.

While the methodological contribution is interesting and promising, the paper would benefit from clearer explanations of architectural components, evaluation variants, and presentation of results. Addressing these clarity issues would significantly strengthen the paper and make its contributions more accessible to the community.

---

### Official Review · Reviewer_Z5F9 · 2026-02-25
**AtomSurf-PPI demonstrates that meaningful protein embeddings, combined with RANSAC, can achieve efficient and accurate protein–protein docking. It challenges the assumption that generative models are necessary, and instead shows that embedding-guided docking can close the gap with traditional methods while outperforming some deep learning baselines.**

**Rating:** 5
**Confidence:** 3

**Review:**

Contributions:

1. Learned protein embeddings themselves are powerful enough to guide docking, without requiring a generative model.
2. Introduces a simple but effective Graph Transformer-based scoring function.
3. Improves deep docking results on the challenging PINDER dataset (Holo, Apo, predicted structures).
4. Leveraging a joint graph + surface encoder to capture both atomic-level graph structure and geometric surface features (a good hybrid alternative to recent generative docking methods)
5. AtomSurf-PPI is strong on experimental structures (Holo/Apo), outperforming AF2-Multimer.

Significance: Shows that embeddings + registration algorithms can deliver state-of-the-art docking without generative modeling.

Cons:

1. Performance collapses on AF2-predicted monomers due to small but systematic errors in surface residue orientations.
2. The rescoring model shows a strong drop between Oracle (ground-truth correspondences) and Top-5 settings indicates that the scoring function is heavily affected by distributional shifts and may not generalize well to noisy or imperfect inputs.
3. No direct comparison with ATOMSURF, despite building on it. An ablation against ATOMSURF would clarify the unique contribution of AtomSurf-PPI beyond the encoder itself.
4. The evaluation could be broadened to include other recent deep docking frameworks (EquiDock etc.)  for a more comprehensive benchmark

---

### Meta-Review · Area_Chair_gPUT · 2026-02-28

**Recommendation:** Accept (Poster)
**Confidence:** 4

**Metareview:**

The reviewers raise good points about clarity and comparisons but this paper is worth discussing at the workshop.

---

### Decision · Program_Chairs · 2026-03-02

**Decision:**

Accept (Poster)

**Comment:**

Please see the meta-review.